# Influence of road environmental factors on traffic accidents involving vulnerable road users through negative binomial models

**Ying Chen, Yi Tian⬥\*, Zhaoheng Ouyang, Jiaxun Zhu**

College of Architecture, Changsha University of Science & Technology, Changsha, Hunan Province, China

\* mowangdal@163.com

**Data Availability Statement:** All relevant data are within the paper and its Supporting Information files.

## Abstract

Ensuring pedestrian safety is crucial for establishing fair and sustainable transportation systems. However, certain demographics face disproportionately higher risks, necessitating age-appropriate policy and design strategies. This study provides a comprehensive analysis of the relationships between objectively measured road infrastructure attributes and pedestrian accident frequencies involving vulnerable groups in Hunan Province, China. By leveraging detailed historical crash records linked to spatially-explicit infrastructure data, the research team employed advanced count regression modeling techniques, including negative binomial (NB) and zero truncated tail negative binomial (ZTNB) specifications, to systematically evaluate the safety impacts of roadway functional classification, intersection design, traffic controls, alignment geometry, pedestrian segregation, land use context, and traffic volumes. The results revealed that the ZTNB approach, which accounted for the excess zero observations inherent to the crash data, provided statistically superior model fit compared to the standard NB formulation. The ZTNB estimation results offered robust empirical evidence regarding key infrastructure risk factors, highlighting that while higher-order roadways exhibited lower pedestrian accident likelihoods, elements such as multi-leg intersections, lack of traffic controls, curved alignments, and absence of segregated facilities correlated with elevated hazards. Older adults and children are particularly susceptible to accidents on major highways and are more prone to traffic incidents on regular roads as opposed to specialized areas like tunnels and intersections. Importantly, the analysis revealed varying safety impacts among different user groups, underscoring the significance of considering the unique requirements and vulnerabilities of diverse pedestrian populations in transportation planning and design. Overall, the findings offer robust empirical evidence to guide development of tailored interventions that consider the unique capacities and exposures of different pedestrian populations. The age-segmented analyses also contribute transportation equity insights for achieving Vision Zero goals through inclusive infrastructure design.

**Funding:** This work was supported in the form of grants from the National Natural Science Foundation of China(No.52102407) awarded to YC. The funders had no role in study design, data collection and analysis, decision to publish, or preparation of the manuscript.https://www.nsfc.gov.cn/.

**Competing interests:** The authors have declared that no competing interests exist.

## Introduction

With the rapid expansion of China's highway network and continuous growth in motor vehicle ownership, the issues of chaotic traffic conditions and traffic accidents have remained persistent challenges. Road traffic injuries pose a significant burden globally, with profound health, social, and economic implications. It is estimated that one person dies in a traffic accident every 24 seconds worldwide [1]. Within this alarming statistic, China has experienced a disproportionately high number and percentage of pedestrian and occupant accidents compared to the total number of traffic crashes (Fig 1). Therefore, comprehending and addressing the factors contributing to pedestrian accidents, particularly among vulnerable populations, is vital from both an equity and public health standpoint.

Specific demographic groups among pedestrians face a significantly elevated risk, notably children and older adults, who confront distinct safety challenges in complex traffic environments due to age-related physical and cognitive limitations. Road traffic injuries are the leading cause of death for children and young people aged 5–29 years [3]. For the elderly population, the mortality rate in road crashes is alarmingly high [4], with the percentage of traffic accident fatalities among the elderly in China reaching 33.6% in 2018 [5]. Rates of pedestrian collisions are disproportionately high among children and older adults compared to the general population, underscoring their heightened vulnerability on roads. Existing research indicates that the physical road environment likely plays a pivotal, albeit intricate, role in determining accident risks for different user groups. Therefore, grappling with such a grave situation, understanding how to effectively mitigate the travel risks faced by these vulnerable groups has become a pressing concern. An essential aspect of this endeavor involves analyzing the role of the road environment in contributing to their traffic accidents [6].

To address these knowledge gaps, the objectives of this study were twofold: 1) to observe the distinct characteristics and differences between the two modeling approaches in capturing

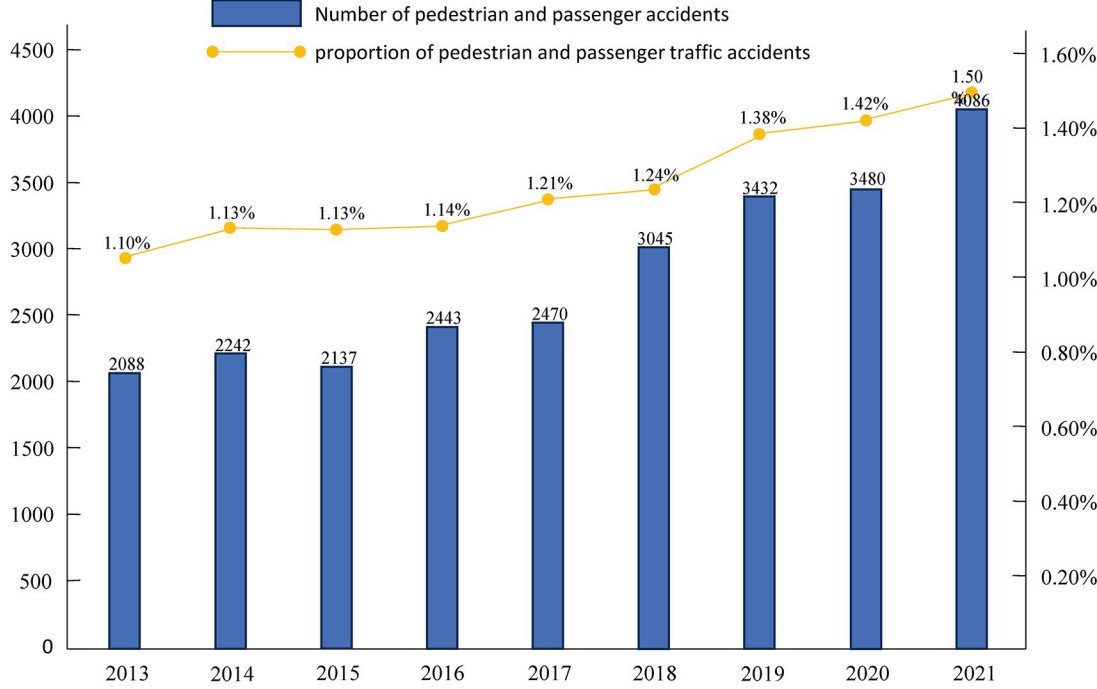

**Fig 1. Number and share of pedestrian-occupant accidents in China's road traffic accidents, 2013–2021 [2].**

the relationship between road environment and traffic accidents, and 2) to derive insights into the travel patterns and safety issues faced by the elderly and young, thereby providing an evidence base to guide optimization of the road environment for improved protection of these vulnerable road users.

Through a comprehensive literature review, this introduction first established the context and rationale for focusing on the traffic safety of elderly and child pedestrians, highlighting the severity of the problem and the critical role of the road environment. It then outlined the specific research goals and modeling techniques employed in the current study. The subsequent sections delved deeper into the data sources, methodological approaches, analysis of results, and implications for enhancing traffic safety for vulnerable populations. Ultimately, this research aimed to contribute to the growing body of knowledge on the complex interplay between road infrastructure design and the safety of pedestrians, particularly those most at risk in the transportation system.

## Literature review

The existing literature has extensively examined the relationship between the road environment and pedestrian traffic safety. Several studies have revealed that changes in the road environment can cause changes in driving safety, changes involving pedestrians are the most hazardous and drivers perceive rural roads to be less hazardous than urban roads [7]. The design of roads and the development of different land uses have been shown to increase or decrease the road traffic injuries suffered by pedestrians [8]. Commercial or high-density residential areas concentrating pedestrian activity tend to see higher crash frequencies, necessitating additional infrastructure supports, yet crash rates per travelled distance may be lower due to traffic calming from mixed usages. Conversely, sparse suburban or rural spacing of destinations discourages walking while facilitating high-speed motorized travel, negatively impacting pedestrian protection.

Researchers have identified various road environmental factors associated with pedestrian crash risk. Zewdie et al. [9] found that pedestrian crashes were linked to road environmental factors such as road density and traffic signals. Xu et al. [10] observed that higher traffic volume was associated with an increased likelihood of crashes, and there were differing associations between land use and crashes in urban and suburban areas. Gálvez-Pérez et al. [11] reported that wider sidewalks and higher signal densities were associated with reduced traffic accidents among elderly pedestrians. Amiour et al. [12] discovered that the presence of sidewalks was linked to the traffic safety of child pedestrians, and intersection density was related to children's safety perceptions. Aceves-González et al. [13] revealed that the absence of traffic signals, excessive traffic, and lack of signs were detrimental to pedestrian traffic safety. T Kraidi et al. [14] identified that both the number of cross-branching roads and cross-traffic volume were significant risk factors associated with pedestrian crash risk.

While effects of design on general pedestrian crash risks are reasonably well-established, research specifically targeting vulnerable elderly and child populations remains limited. Early studies on vulnerable groups, specifically the elderly and children, predominantly examined their behavior, environment, and other research starting points. Studies have revealed that in traffic accidents, the elderly are more likely to sustain injuries to the torso, pelvis, and limbs, with a lower incidence of head injuries, and exhibit a higher mortality rate in hospitals [15]. Ang et al. [16], in his investigation into the causes of death in road traffic accidents among the elderly, noted a higher mortality rate among pedestrians aged over 75 years. Ren et al. [17], in their controlled experiment comparing the mobility of the elderly with that of the young, determined that the extended time spent by the elderly at crossings was a primary factor

contributing to their decreased mobility. Laković et al. [18], in their research, highlighted that suboptimal external conditions for older adults may expose them to more hazardous situations, potentially leading to misbehavior.

Regarding child pedestrians, studies have shown that less urbanized counties are associated with higher fatality rates in their examination of child fatalities from road traffic crashes in the United States [19]. Chong et al. [20], in their research, highlighted that children are particularly susceptible to severe injuries when traveling as pedestrians. Wang et al. [21] identified that children's emotional fear could predict risk-taking behavior in traffic, with fear potentially leading to excessive caution and inefficiency in less risky traffic conditions. Zhao et al. [22] determined that the pedestrian accessibility and safety of children's school travel routes are influenced by factors such as effective walking width, spatial connectivity, visual integration, pedestrian safety obstacles, cross-facility completeness, and traffic flow.

More recent analyses have employed advanced statistical techniques to disentangle environmental influences from other determinants. Negative binomial (NB) regression allows controlling for confounding factors and exposure while detecting associations between attribute levels and injury counts. Zero truncated tail negative binomial (ZTNB) models further address issues of excess zeros in safety datasets and improve fitting for crash frequency analysis. By rigorously modeling the statistical properties of accident data, these analytical approaches can provide robust, data-driven evidence to guide targeted infrastructure interventions and policy decisions.

In summary, the extant literature highlights the versatility and power of count-based regression techniques, particularly the NB and ZTNB models, in uncovering the complex interplay between transportation infrastructure, land use, and the safety of vulnerable road users. However, the specific application of these methodologies to pedestrian safety issues in the Chinese context, with a focus on high-risk elderly and child populations, remains an underexplored area warranting further investigation. Therefore, this study aimed to address this gap by using historical traffic accident data from Hunan Province, China, to examine the relationship between the road environment and the traffic safety of the elderly and young groups.

## Data and methodology

### Study area and data sources

This study utilized traffic accident data collected by the Hunan Provincial Public Security Department in China, covering the 3-year period from 2014 to 2016. Hunan is a populous inland province located in south-central China, with a total area of around 210,000 square kilometers and a permanent residential population of over 65 million as of 2020. The provincial capital Changsha and several other major cities in Hunan have experienced rapid urbanization and motorization in recent decades, leading to rising concerns over traffic safety, especially for vulnerable road users.

The accident dataset contained detailed information on each reported collision, including the date, time, location, vehicle types involved, and injury severity outcomes. For the current analysis, only records pertained to pedestrian accidents were extracted. The geocoded spatial coordinates provided in the dataset allowed linking each crash observation to corresponding roadway segments using GIS mapping software. This level of granularity in the data allows for a rigorous, quantitative examination of the influence of specific road environment factors on pedestrian accident frequencies.

Additionally, the age of persons involved in crashes was available, permitting stratification into vulnerable subgroups. Based on the data source the target population selected for this study is the elderly and children among the vulnerable groups on the road. Specifically,

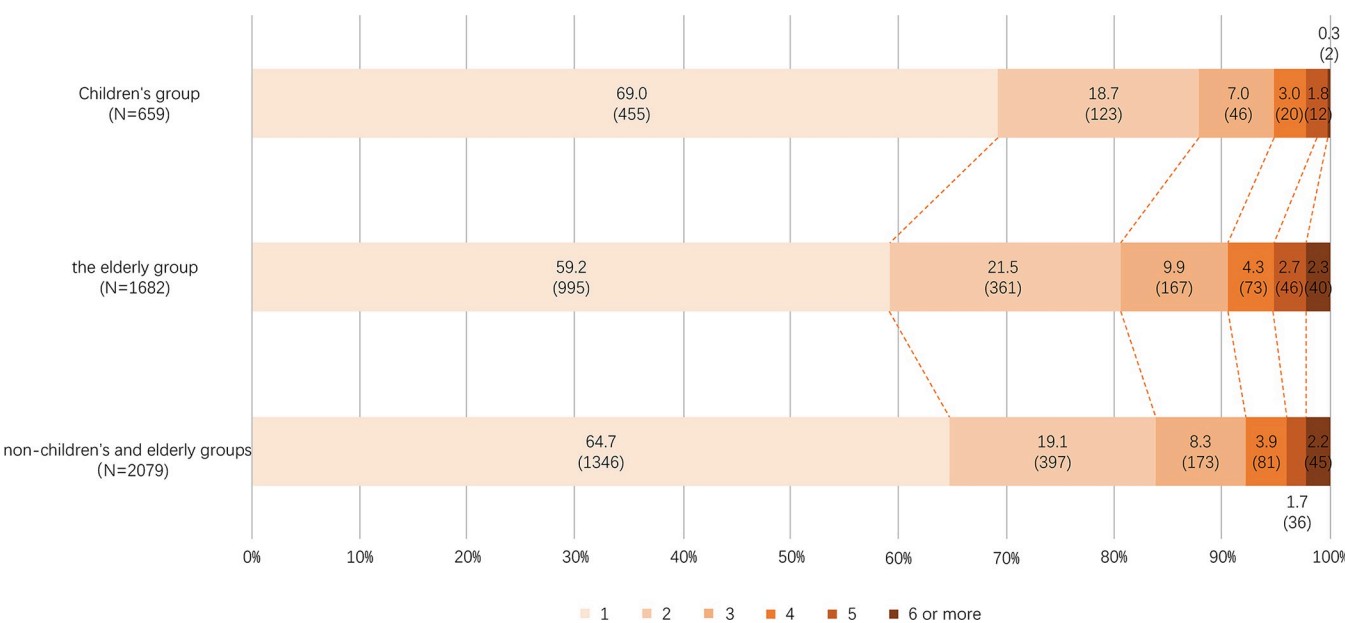

**Fig 2. Comparison of the data structures of the three groups.**

accidents involving pedestrians aged 60 years or above were categorized under the 'Elderly' group, while those relating to individuals between 1–12 years comprised the 'Child' group. All remaining observations not meeting these age criteria fell under a 'General' population category to serve as a baseline control group for comparative modeling.

After conducting data screening, which included missing data imputation and address geocoding, the final sample sizes comprised 1,682 accident records for the Elderly group, 659 records for Children, and 2,079 records representing the General pedestrians. Anomalous data can lead to misinterpretation when analyzing the results of fatal urban traffic accidents. Therefore, both the detection of abnormal data and the recovery of missing data are crucial in data preprocessing [23]. Traffic accident data from the entire year 2014–2016 in Hunan Province, China, underwent initial screening to replace missing values with the mean or median [24]. Due to the specificity of traffic accident statistics, the data was obtained from incidents with several accidents greater than zero (zero-truncated data). Subsequently, the accident locations were categorized based on road sections, and incidents occurring in the same road sections were aggregated. This process yielded a total of 1,682 groups of traffic accident data for the elderly, 659 groups for children, and 2,079 groups for non-children and elderly groups. The three data groups are structured as depicted in Fig 2.

## Environmental variables

Based on prior literature substantiating their relevance to pedestrian safety and crash mechanisms, a set of roadway attributes reflecting urban form, traffic operations, and user separation characteristics were selected for systematic data collection and modeling. These factors were measured objectively along road segments corresponding to accident locations to characterize local infrastructural conditions at the time of each crash event.

Data on the accident location, road safety control, road alignment, intersection section type, road class, isolation type for classification and assignment of values were manually extracted from high-resolution satellite imagery and validated using online street view

**Table 1. Road environment impact factors.**

| Impact Factors | Description | Value Assignment |
|---|---|---|
| Location of the accident | Sidewalks | 1 |
|  | Pedestrian crossing | 2 |
|  | Nonmotorized road | 3 |
|  | Nonmotorized Mixed Lane | 4 |
|  | motorway | 5 |
| Road safety control | Composite safety control | 1 |
|  | Car indicator | 2 |
|  | Other security facilities | 3 |
|  | Traffic marking | 4 |
|  | Traffic sign | 5 |
|  | Uncontrolled | 6 |
| Road alignment | Straight | 1 |
|  | Winding road | 2 |
|  | Ramp | 3 |
|  | Winding slope | 4 |
| Type of intersection section | Ordinary road | 1 |
|  | Narrow road | 2 |
|  | Access to the roadway | 3 |
|  | Branch port | 4 |
|  | Other special road sections | 5 |
| Road class | Freeway | 1 |
|  | City road | 2 |
|  | Other roads | 3 |
| Isolation type | Segregation Type 1 | 1 |
|  | Segregation Type 2 | 2 |
|  | Unisolated | 3 |

resources where available. Additionally, descriptive statistics of traffic volumes obtained from municipal count stations were joined spatially using GIS techniques.

Table 1 summarizes the environmental variables considered in the analysis and their coding schema, reflecting commonly adopted taxonomies in international road safety literature as well as China's national design guidelines. Composite safety control in road safety management refers to road control measures that incorporate two or more road control methods. Other safety facilities include lighting equipment, contour markers, guide signs, anti-glare installations, and additional safety measures. Special road sections within the intersection category refer to unique configurations such as tunnels, bridges, hazardous roadside areas, elevated structures, and roundabout intersections. Urban roads encompass first, second-, third-, and fourth-class roads, while "others" denotes roads that do not meet the minimum standards of the lowest functional class of highways. Segregation Type 1 indicates the presence of a central barrier with a non-motorized barrier, whereas Segregation Type 2 signifies the existence of either a central barrier or a non-motorized barrier alone.

## Modeling approach

To quantify the relationships between the road environment characteristics and pedestrian accident frequencies, this study employed two distinct count-based regression modeling techniques: the NB model and the ZTNB model.

## Negative binomial model

The NB model is a popular analytical tool in traffic safety [25]. The NB model is based on the prior Poisson prediction model, which is proposed when the Poisson prediction model is over- or under-discretized (fails to meet its fundamental assumptions that the mean and variance are equal). The NB model also follows the shape of the Poisson distribution, which is as follows:

$$P(Y|\mu) = \frac{\mu^Y \times e^{-\mu}}{Y!} \qquad Y = 0, 1, 2, \cdots \tag{1}$$

where $Y$ represents the number of accidents occurring and μ the expected value of the number of accidents occurring, which can also be considered as a function that conforms to the conditions related to the impact of accidents, if the mean value $\mu$ is related to the factors influencing the accidents.

$$\mu = f(\text{Road impact factors}) \tag{2}$$

When the data is taken to a non-zero value, the conditional probability in Eq (1) is

$$P(Y|Y > 0) = \frac{P(Y)}{P(Y > 0)} = \frac{P(Y)}{1 - P(Y = 0)} \qquad Y = 0, 1, 2, \cdots \tag{3}$$

## Zero truncated tail negative binomial models

Traditional negative binomial regression models face limitations when analyzing traffic accident count data that only includes reported, non-zero events due to data collection methods. This renders it difficult to accurately model the full phenomenon including zero-occurrence observations that are unrecorded. To address this, a ZTNB model was developed based on the standard negative binomial framework to more precisely capture the statistical attributes of such datasets. The ZTNB relaxes two key assumptions of the negative binomial–that $\mu$ follows a gamma distribution, and $\alpha$ represents the discrete coefficient within that distribution.

By accounting for the inherent zero-truncation of only recording non-zero crash events, Chen et al. [26, 27] derived the probability mass function of the zero-truncated variable as: Assuming $\mu$ obeys the Gamma distribution, where $\alpha$ is the discrete coefficient in the Gamma distribution, the distribution probability function of the zero-truncated-tailed variable is finally obtained in Eq (4) and Eq (5). This modified formulation provides a theoretically sounder approach for modeling traffic accident counts by acknowledging the data properties introduce biases when directly applying traditional count techniques.

$$P(Y = 0|\mu, \alpha) = \frac{\Gamma(y + \alpha)}{\Gamma(Y + 1) \times \Gamma(\alpha)} \times \left(\frac{\mu/\alpha}{1 + \left(\frac{\mu}{\alpha}\right)}\right)^Y \times \left(\frac{1}{1 + \left(\frac{\mu}{\alpha}\right)}\right)^\alpha = \left(\frac{1}{1 + \frac{\mu}{\alpha}}\right)^\alpha \tag{4}$$

$$P(Y|\mu, \alpha, Y > 0) = \frac{\Gamma(y + \alpha)}{\Gamma(Y + 1) \times \Gamma(\alpha)} \times \left(\frac{\mu/\alpha}{1 + \left(\frac{\mu}{\alpha}\right)}\right)^Y \times \left(\frac{1}{1 + \left(\frac{\mu}{\alpha}\right)}\right)^a / \left(1 - \left(\frac{1}{1 + \frac{\mu}{\alpha}}\right)^\alpha\right) \tag{5}$$

where $\Gamma$ is the gamma distribution, $Y$ is the number of vehicle accidents, and $\mu$ is the mean of the number of accidents.

In the NB model and the ZTNB model, after assuming that there is a relationship between the number of accidents and the remaining accident elements, the relationship between μ and

the remaining accident elements is:

$$\log(\mu) = \beta_1 + \sum \beta_i \log(x_i) \tag{6}$$

where $\beta_1$ is the coefficient of the intercept, $x_i$ is the assigned value of the various accident elements, and $\beta_i$ is the expected corresponding coefficient. The solution of the NB model with ZTNB model is generally solved for the parameters using maximum likelihood estimation.

The ZTNB model possesses a key advantage over the traditional NB model in its ability to account for the nature of traffic accident count data that is inherently zero-truncated. Specifically, real-world collision datasets only record incidents where harm occurred, rather than the full spectrum of occurrences and non-occurrences. The ZTNB addresses this limitation by incorporating an additional parameter to explicitly model the probability of a non-zero, observed event. This directly acknowledges the data do not capture the complete theoretical distribution including zero values. By relaxing the NB assumption of a singular probability function, the ZTNB is better positioned to yield accurate maximum likelihood estimates of regression coefficients through differentiation of the truncated and non-truncated components.

The ZTNB model is estimated using maximum likelihood techniques, with the log-likelihood function modified to account for the truncation at zero. The resulting coefficient estimates can be interpreted similarly to the NB model, representing the expected change in the log of the non-zero accident count associated with a unit change in the corresponding explanatory variable.

## Comparison and analysis of results

### Comparison of results for the elderly group

The data structure and model results for the elderly group are presented in Tables 2 and 3. The NB model and ZTNB model both identified several significant factors related to the number of traffic accidents among the elderly.

The most influential factor was road class, with larger negative coefficients in both models (-0.1440 and -0.3286, respectively), indicating that as the road class increases (e.g., from lower-class urban roads to highways), the number of accidents involving the elderly tends to decrease. This suggests that high-class roads may provide a safer environment for elderly pedestrians compared to lower-class urban roads. Other significant factors include accident location, road safety control, and intersection section type. The positive coefficient for accident location indicates that accidents are more likely to occur in more dangerous road sections, such as on the roadway rather than on sidewalks. The negative coefficients for road safety control and intersection section type suggest that better safety measures and less complex intersections are associated with fewer accidents involving the elderly. In contrast,

**Table 2. Data structure of the elderly group.**

| Parameter | Maximum Values | Minimum Value | Average Value |
|---|---|---|---|
| Number of accidents | 6.000 | 1.000 | 1.772 |
| Location of the accident | 5.000 | 1.000 | 4.357 |
| Road safety control | 6.000 | 1.000 | 3.891 |
| Road alignment | 4.000 | 1.000 | 1.424 |
| Type of intersection section | 5.000 | 1.000 | 1.887 |
| Road class | 3.000 | 1.000 | 2.096 |
| Isolation type | 3.000 | 1.000 | 2.770 |

**Table 3. Results of NB and ZTNB models for the elderly group.**

| Parameter | NB Model | | | | ZTNB Model | | | |
|---|---|---|---|---|---|---|---|---|
| | Estimated | Standard deviation | P-value | Significance | Estimated | Standard deviation | P-value | Significance |
| $\beta_1$ | 0.7898 | 0.1619 | 1.07e-06 | 99.9% | 0.1044 | 0.3703 | 0.7779 | Not significant |
| $\beta_2$ | —— | —— | —— | —— | -0.3563 | 0.2634 | 0.1760 | Not significant |
| $\beta_{LOC}$ | 0.0495 | 0.0186 | 0.0079 | 99.0% | 0.1184 | 0.0408 | 0.0037 | 99.0% |
| $\beta_{CON}$ | -0.0263 | 0.0093 | 0.0047 | 99.0% | -0.0642 | 0.0201 | 0.0013 | 99.0% |
| $\beta_{ALI}$ | -0.0066 | 0.0230 | 0.7712 | Not significant | -0.0074 | 0.0495 | 0.8802 | Not significant |
| $\beta_{INT}$ | -0.0444 | 0.0133 | 0.0008 | 99.9% | -0.1059 | 0.0289 | 0.0002 | 99.9% |
| $\beta_{ROA}$ | -0.1440 | 0.0497 | 0.0037 | 99.0% | -0.3286 | 0.1082 | 0.0023 | 99.0% |
| $\beta_{ISO}$ | 0.0206 | 0.0434 | 0.6337 | Not significant | 0.0256 | 0.0929 | 0.7824 | Not significant |
| AIC | 5056.16 | —— | —— | —— | 4042.12 | —— | —— | —— |
| BIC | 5099.58 | —— | —— | —— | 4085.54 | —— | —— | —— |
| MSE | 1.3947 | —— | —— | —— | 1.3943 | —— | —— | —— |
| RMSE | 1.1810 | —— | —— | —— | 1.1808 | —— | —— | —— |

road alignment and segregation type did not show significant relationships with accident frequency in the elderly group. This implies that factors like road curvature and the presence of physical barriers may not be the primary determinants of traffic safety for elderly pedestrians in this context.

## Comparison of results for the children's group

The data for the children's group were also analyzed using the NB and ZTNB models to investigate the relationship between road environment and traffic accident frequency. The data structure for the children's group and the model results are presented in Tables 4 and 5, respectively.

In the two models for the children's group, significant factors influencing the number of traffic accidents included intersection section type and road grade. Notably, the estimated coefficient for road grade exhibited the most pronounced impact among the significant factors. Specifically, the estimated coefficients for road grade in both models were -0.1920 and -0.7381, respectively, indicating that road grade exerted the most substantial influence on the number of traffic accidents involving children at the same intersection section. Moreover, the analysis revealed a positive correlation between higher road grades and increased accident rates for children. Interestingly, the occurrence of accidents involving children was found to be lower in more specialized road sections. Conversely, other factors within the children's

**Table 4. Data structure of the children's group.**

| Parameter | Maximum Values | Minimum Value | Average Value |
|---|---|---|---|
| Number of accidents | 8.000 | 1.000 | 1.516 |
| Location of the accident | 5.000 | 1.000 | 4.332 |
| Road safety control | 6.000 | 1.000 | 4.930 |
| road alignment | 4.000 | 1.000 | 1.458 |
| Type of intersection section | 5.000 | 1.000 | 1.624 |
| road class | 4.000 | 1.000 | 2.164 |
| Isolation type | 3.000 | 1.000 | 2.906 |

**Table 5. Results of NB and ZTNB models for the children's group.**

| Parameter | NB Model | | | | ZTNB Model | | | |
|---|---|---|---|---|---|---|---|---|
| | Estimated | Standard Deviation | P-Value | Significance | Estimated | Standard Deviation | P-Value | Significance |
| $\beta_1$ | 0.8488 | 0.3394 | 0.0124 | 95.0% | 0.5954 | 0.9128 | 0.5142 | Not significant |
| $\beta_2$ | —— | —— | —— | —— | -0.8742 | 0.7186 | 0.2238 | Not significant |
| $\beta_{LOC}$ | 0.0129 | 0.0291 | 0.6561 | Not significant | 0.0308 | 0.0700 | 0.6598 | Not significant |
| $\beta_{CON}$ | -0.0317 | 0.0188 | 0.0923 | Not significant | -0.0840 | 0.0444 | 0.0586 | Not significant |
| $\beta_{ALI}$ | 0.0296 | 0.0410 | 0.4708 | Not significant | 0.0735 | 0.0979 | 0.4527 | Not significant |
| $\beta_{INT}$ | -0.0620 | 0.0269 | 0.0213 | 95.0% | -0.1975 | 0.0676 | 0.0035 | 99.0% |
| $\beta_{ROA}$ | -0.1920 | 0.0883 | 0.0298 | 95.0% | -0.7381 | 0.2278 | 0.0012 | 99.0% |
| $\beta_{ISO}$ | 0.0457 | 0.1017 | 0.6533 | Not significant | 0.0976 | 0.2372 | 0.6806 | Not significant |
| AIC | 1792.57 | —— | —— | —— | 1264.07 | —— | —— | —— |
| BIC | 1828.49 | —— | —— | —— | 1299.99 | —— | —— | —— |
| MSE | 0.8781 | —— | —— | —— | 0.8781 | —— | —— | —— |
| RMSE | 0.9371 | —— | —— | —— | 0.9365 | —— | —— | —— |

group data, such as accident location, road safety control, road alignment, and segregation type, did not exhibit statistical significance in these two models.

## Comparative analysis of the non-children and elderly group results

To determine the characteristics of pedestrian traffic accidents for the young and old groups, the non-children and elderly groups were used as a control group. Two models were fitted using the data structure for the non-children and elderly groups (Table 6), and the results are presented in Table 7.

In both models for the non-children and elderly groups, road traffic control and road segregation were strongly associated with the number of traffic accidents. The type of segregation was the most significant factor, with estimated coefficients of -0.1384 and -0.2937 in the two models, respectively. This indicates that the more comprehensive the segregation, the fewer the locations where traffic accidents occurred. Additionally, the better the traffic regulation, the more sites where there were traffic accidents. Other characteristics, including accident location, intersection section type, road class, and road alignment, were not found to be significant in either model for the non-children and elderly groups.

The comparative analysis of the results across the different groups provides insights into the distinct relationships between road environment factors and traffic accident frequency for the elderly, children, and the non-children and elderly groups. These findings can inform the development of targeted interventions and the optimization of road environments to enhance the safety of vulnerable road users.

**Table 6. Data structure of the non-children and elderly groups.**

| Parameter | Maximum Values | Minimum Value | Average Value |
|---|---|---|---|
| Number of accidents | 9.000 | 1.000 | 1.669 |
| Location of the accident | 5.000 | 1.000 | 4.574 |
| Road safety control | 6.000 | 1.000 | 4.930 |
| Road alignment | 4.000 | 1.000 | 1.363 |
| Type of intersection section | 5.000 | 1.000 | 1.846 |
| Road class | 4.000 | 1.000 | 2.047 |
| Isolation type | 3.000 | 1.000 | 2.734 |

**Table 7. Results of NB and ZTNB models for the non-children and elderly groups.**

| Parameter | NB Model | | | | ZTNB Model | | | |
|---|---|---|---|---|---|---|---|---|
| | Estimated | Standard deviation | P-value | Significance | Estimated | Standard deviation | P-value | Significance |
| $\beta_1$ | 1.1358 | 0.1574 | 5.49e-13 | 99.9% | -0.1548 | 0.7306 | 0.8321 | Not significant |
| $\beta_2$ | —— | —— | —— | —— | -2.0233 | 0.7478 | 0.0068 | 99.0% |
| $\beta_{LOC}$ | -0.0169 | 0.0207 | 0.4141 | Not significant | -0.0257 | 0.0541 | 0.6345 | Not significant |
| $\beta_{CON}$ | -0.0334 | 0.0085 | 8.86e-05 | 99.9% | -0.0933 | 0.0218 | 1.94e-05 | 99.9% |
| $\beta_{ALI}$ | -0.0270 | 0.0244 | 0.2679 | Not significant | -0.0580 | 0.0622 | 0.3507 | Not significant |
| $\beta_{INT}$ | -0.0042 | 0.012 | 0.7246 | Not significant | -0.0070 | 0.0310 | 0.8212 | Not significant |
| $\beta_{ROA}$ | -0.0045 | 0.0443 | 0.9189 | Not significant | -0.1301 | 0.1151 | 0.2582 | Not significant |
| $\beta_{ISO}$ | -0.1384 | 0.0362 | 0.0001 | 99.9% | -0.2937 | 0.0957 | 0.0021 | 99.0% |
| AIC | 6111.97 | —— | —— | —— | 4582.87 | —— | —— | —— |
| BIC | 6157.09 | —— | —— | —— | 4627.98 | —— | —— | —— |
| MSE | 1.40 | —— | —— | —— | 1.40 | —— | —— | —— |
| RMSE | 1.18 | —— | —— | —— | 1.18 | —— | —— | —— |

## Discussion

### Comparative performance of modeling approaches

This study applied two count regression modeling techniques—the NB model and the ZTNB model—to examine the relationships between road environment characteristics and pedestrian accident frequencies among different groups in Hunan Province, China. A comparative analysis of the modeling results provides useful insights into the performance of these approaches.

In comparing the strengths and weaknesses of the models, the size of the Akaike Information Criterion (AIC) and Bayesian Information Criterion (BIC) values is commonly utilized, where smaller values indicate better predictive and fitting abilities of the models. As shown in Table 8, the AIC and BIC values of the ZTNB model are notably smaller than those of the NB model. Additionally, while the mean squared error (MSE) and root mean squared error (RMSE) were similar between the two approaches, the ZTNB specification still demonstrated marginally superior predictive accuracy.

Overall, the comparative assessment confirms that accounting for dataset properties through the ZTNB approach provides a more refined, accurate representation of complex safety phenomena compared to standard NB regression. This systematic comparison validates the zero-truncated model as the preferred analytical framework for the current study's intended purposes.

**Table 8. Comparing the outcomes of the three groups of models.**

| | Children's Group | | Elderly Group | | Non-children Group | |
|---|---|---|---|---|---|---|
| | NB model | ZTNB model | NB model | ZTNB model | NB model | ZTNB model |
| Location of the accident | Not significant | Not significant | 99.0% | 99.0% | Not significant | Not significant |
| Road safety control | Not significant | Not significant | 99.0% | 99.0% | 99.9% | 99.9% |
| Type of intersection section | 95.0% | 95.0% | 99.9% | 99.9% | Not significant | Not significant |
| Road class | 95.0% | 95.0% | 99.0% | 99.0% | Not significant | Not significant |
| Isolation type | Not significant | Not significant | Not significant | Not significant | 99.9% | 99.0% |
| Aic | 1792.57 | 1264.07 | 5056.16 | 4042.12 | 6111.97 | 4582.87 |
| Bic | 1828.49 | 1299.99 | 5099.58 | 4085.54 | 6157.09 | 4627.98 |
| Mse | 0.87 | 0.87 | 1.39 | 1.39 | 1.40 | 1.40 |
| Rmse | 0.93 | 0.93 | 1.18 | 1.18 | 1.18 | 1.18 |

## Key findings and vulnerable group insights

Beyond the advantages of the ZTNB specification demonstrated above, several substantive findings emerge from the quantitative analysis that provide useful insights for decision makers. Consistent with previous literature, infrastructure elements like functional classification, intersection design, traffic controls, geometry and segmentation were clearly associated with pedestrian accident likelihoods. However, certain attributes differed substantially in their effects depending on the vulnerable group.

In comparison with previous studies, it was observed in our research that elderly individuals tended to select shorter time intervals, maintain closer proximity, exhibit increased cooperation, and experience fewer missed opportunities when confronted with higher vehicle speeds at intersections, consequently leading to a higher incidence of traffic accidents [28]. Consistent with the findings of Lee S et al. [29], our conclusions align with the impact of road section types at intersections. Ordinary four-way intersections were identified as potential sites of numerous traffic conflict points between pedestrians and vehicles, with elderly individuals showing a lower likelihood of being involved in accidents on specialized road sections. Regarding signal control, a study by Ma CX et al. [30] revealed a gradual increase in the proportion of middle-aged and elderly pedestrians crossing roads at signalized intersections, engaging in behaviors that posed potential risks, a trend that corresponds with the outcomes of our study. Furthermore, the research conducted by Zhang HL et al. [31] supported our conclusions by emphasizing that elderly pedestrians, when compared to younger individuals, exhibited decreased overall perceptual efficiency, shorter gaze duration, and increased cognitive load at signalized intersections. Regarding segregation facilities, findings from Wu H et al. [32] suggested that the presence of shrubs could obstruct the visibility of pedestrians and vehicle drivers, thereby increasing the likelihood of collisions. However, our analysis did not reveal a clear correlation between the presence of shrubs and the occurrence of accidents.

The comparison of the parameter sizes for the three sets of influencing factors (Fig 3) provides additional insights:

Road grade is the most influential factor on the number of accidents in both young and elderly groups, however, the higher the road grade often means greater traffic flow and speed, the young and elderly groups of vehicle speed, trajectory judgment ability is weak, unable to grasp the complexity of the fast traffic scene. In the road environment, as the road level increases, clear and easy-to-understand traffic signs should be increased, while the accident-prone road in the case of the impact of driving safety, speed limits, speed bumps and other measures can be used appropriately to reduce the occurrence of traffic accidents.

Both young and elderly groups demonstrate that the number of accidents in more special junctions is lower, whereas the number of accidents in conventional crossroads is higher. First and foremost, the number of special junctions is lower than the number of regular crossroads, and unique intersections such as tunnels and bridges are easy to notice, increasing people's vigilance. This minimizes the likelihood of crossing unique road sections such as three-way junctions, resulting in fewer accidents [33]. Although the type of segregation does not show significant characteristics, some higher protection zones and barriers can be used to prevent pedestrians from entering or performing dangerous crossing behaviors on special road sections. Second, the bigger the number of accidents on common routes, the more likely it is that the elderly and young are limited by their own reasons for lacking transportation safety awareness. Due to a lack of experience, children are frequently irresponsible and hasty, and they lack basic transportation awareness [34]. Elderlyer pedestrians who use traditional mobility aids such as canes, walkers, and wheelchairs daily are more likely to be involved in pedestrian accidents [35]. In some areas where there are large groups of young and elderly group, traffic

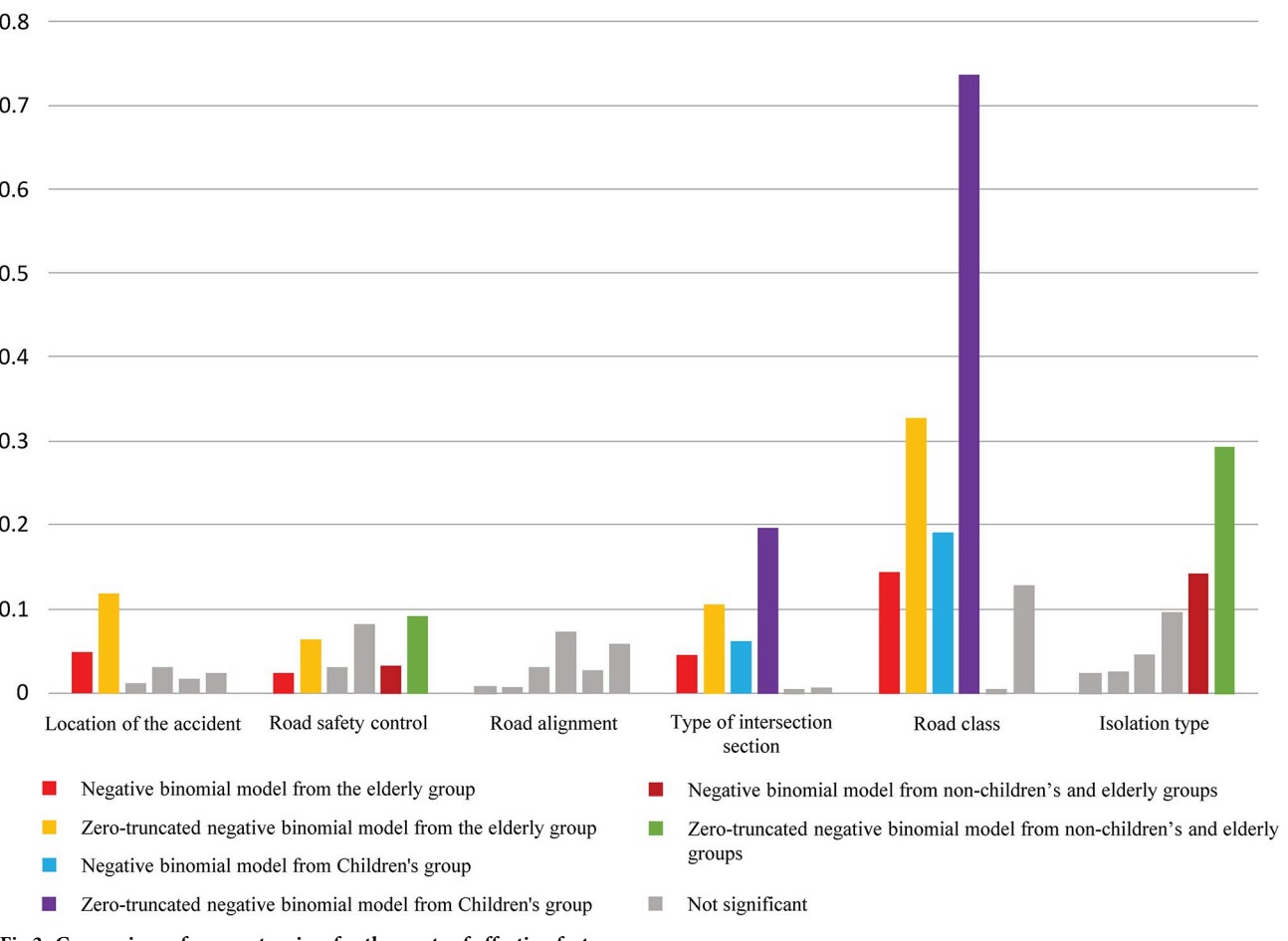

**Fig 3. Comparison of parameter sizes for three sets of affecting factors.**

signals can be adjusted to allow enough time for them to pass, or in the roadway to reduce the number of accidents involving young and elderly groups by increasing the number of traffic police officers directing the roadway and guiding the roadway by a special person at the intersection.

Accident location is a road element that primarily impacts elderlyer age groups. Accidents are more common on dangerous roads, such as motorways and motorized/non-motorized roads. The primary cause for this is the limitations of elderly people's own variables and their reduced capacity to regulate traffic situations. Younger pedestrians cross the street more safely because they pay closer attention to motor traffic, but senior pedestrians pay more attention to the street since they must plan their steps better [36]. When crossing the street, elderly pedestrians turn their heads less frequently, and they estimate the distance to approaching automobiles less accurately [37]. It is probable that as people age, their judgment of road conditions deteriorates, and they fail to comply with traffic restrictions, putting them in unsafe situations where accidents are common. Separation zones and protection zones on roadways are still effective strategies to keep the elderly out of dangerous locations, and the proper placement of pedestrian islands in the middle of the road will make it much easier for the slow-moving elderly to cross. Where there is a greater need to cross, pedestrian crossings, bridges, or underpasses can be built to separate the elderly from automobiles and lessen the likelihood of an accident.

In terms of traffic control, our study revealed that both elderly and non-elderly individuals were more prone to accidents in areas with enhanced traffic management features, such as

traffic signals, signs, and markings, as compared to findings from other research [18]. In regions boasting well-developed transportation infrastructure, the occurrence of certain violations may have adverse effects, potentially linked to age-related factors. Older individuals often exhibit a tendency to disregard their right of way and possess limited foresight regarding vehicle movements, thereby increasing the likelihood of accidents. In contrast, younger individuals tend to utilize technological devices that divert their attention [38]. To address areas with a high risk of accidents, strategies such as adjusting signal timings to extend pedestrian crossing durations without impacting traffic flow, altering traffic control mechanisms (e.g., introducing artificial traffic controls), and installing additional signs in accident-prone locations to encourage vehicle deceleration have been proposed.

In the case of the barrier, its existence makes it impossible for vehicles to move outside the regular travel area, and violations such as crossing the barrier by non-elderly or young individuals enhance the likelihood of an accident. In the parameters used, there is no significant relationship between road linearity and traffic accidents. In terms of road linearity, the presence of curves, up and down slopes reduce the speed of the vehicle due to the driver's initiative, and in special road linearity, people concentrate their attention, reducing the likelihood of accidents, contrary to our subconscious belief that curves and slopes have a very strong relationship with the occurrence of accidents.

Further analysis of the data revealed that the positive and negative signs of the estimated coefficients of the two model comparisons within the same group were consistent, but the magnitude of the values varied, with the estimated coefficients in the ZTNB model all being larger, indicating a greater influence on the dependent variable. In the comparison of the same models in three groups, the positive and negative sign of the significant factors did not change, indicating that there is indeed a potentially influential relationship between the influencing factors and the number of accidents in the accident data, and that the magnitude and significance of the influences are not the same in the different models.

## Limitations and future research

Due to some legal and regulatory restrictions, accident data for nearby years cannot be obtained. This does not prevent us from identifying some accident characteristics of the elderly and young population from accident data over the past 10 years, but we still hope to obtain the latest data to make our research more comprehensive.

Several limitations remain that future work could help address. First, unobserved confounding by unmeasured risk factors possibly associated with the infrastructure characteristics could bias results to an unknown extent. However, controlling for many road environment covariates aimed to minimize this threat to internal validity.

Second, the generalizability of insights beyond the specific study context of Hunan Province, China, with unique infrastructure standards and travel patterns, requires cautious interpretation and validation elsewhere. Nevertheless, enhancing road safety science globally relies on evidence accretion from diverse transportation settings.

Third, the cross-sectional study design precludes conclusions regarding causal directionality between variables. Extending models longitudinally could help establish temporality and monitor safety impacts of infrastructure modifications over time. Additionally, jointly modeling crash frequencies and severity outcomes may further disentangle risk factor relationships.

## Conclusions

This study employed rigorous count-based regression modeling techniques to disentangle the complex relationship between the road environment and traffic accident risks faced by

vulnerable road user groups, including the elderly and children, in China's Hunan Province. The application of both NB and ZTNB models allowed for robust examination of the factors influencing pedestrian accident frequencies while accounting for the unique statistical properties of the accident data.

The analysis revealed distinct patterns in how the road environment shapes safety outcomes for the elderly versus children. For the elderly group, key influential factors included road class, accident location, road safety control measures, and intersection configurations. Specifically, the results indicated that higher-class roads, improved safety infrastructure, and simpler intersections were associated with lower accident frequencies among elderly pedestrians. In contrast, the child pedestrian group was found to be more sensitive to road grade, with steeper gradients significantly increasing their crash risk. Intersection complexity also emerged as a notable determinant for child traffic safety.

These divergent findings underscore the need for tailored, age-segmented approaches to transportation planning and infrastructure design. Measures to enhance the mobility and safety of the elderly, such as expanding accessibility on urban streets, upgrading crossing facilities, and implementing traffic calming features, may be crucial. For child pedestrians, strategies focused on reducing speed differentials, improving crossing opportunities, and strengthening separation between motorized and non-motorized modes could yield substantial safety benefits.

The study results suggest that to enhance safety for the elderly and young individuals on major roadways, changes need to be made to the traffic environment. Strategies such as increasing the number of signs, reinforcing existing ones, or implementing traffic police patrols to alert people to potential dangers and discourage them from accessing higher-level roadways are recommended. Recommendations for intersections include the installation of pedestrian bridges in busy areas to reduce accident rates, the addition of pedestrian islands or sidewalks, and the expansion of refuge areas in the middle of the road to prevent individuals from crossing or passing through from a safe location. These measures are aimed at preventing accidents involving young people and the elderly on ordinary road sections.

By providing robust, data-driven evidence on the nuanced relationships between the road environment and the safety of vulnerable road users, this study contributes to the growing literature on transportation equity and sustainable mobility. The insights gained can inform the development of more inclusive and age-friendly transportation systems, ultimately advancing the goal of Vision Zero and ensuring safe, accessible journeys for all members of the community.

## Supporting information

**S1 File. The support files include the children group.csv and the elderly group.csv, which are categorised according to Table 1.** The num in the data represents the number of accidents at the location; the hdm represents the accident location; the con represents the road safety control; the roadty1 represents the road alignment; the roadty2 represents the type of intersection section; the roadty3 represents the road class; and the iso represents the type of segregation.
(ZIP)

## Author Contributions

**Conceptualization:** Ying Chen.

**Data curation:** Ying Chen, Yi Tian.

**Formal analysis:** Ying Chen, Yi Tian.

**Funding acquisition:** Ying Chen.

**Investigation:** Zhaoheng Ouyang, Jiaxun Zhu.

**Methodology:** Ying Chen.

**Project administration:** Ying Chen.

**Resources:** Ying Chen.

**Software:** Ying Chen.

**Supervision:** Ying Chen.

**Validation:** Zhaoheng Ouyang, Jiaxun Zhu.

**Visualization:** Yi Tian.

**Writing – original draft:** Yi Tian.

**Writing – review & editing:** Yi Tian.

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
