## [Decision Letter · Decision Letter 0]

24 Sep 2024

PONE-D-24-31325Influence of Road Environmental Factors on Traffic Accidents Involving Vulnerable Road Users through Negative Binomial ModelsPLOS ONE

Dear Dr.Tian,

Thank you for submitting your manuscript to PLOS ONE. After careful consideration, we feel that it has merit but does not fully meet PLOS ONE’s publication criteria as it currently stands. Therefore, we invite you to submit a revised version of the manuscript that addresses the points raised during the review process.

Please, consider all comments

We look forward to receiving your revised manuscript.

Kind regards,

Ahmed Mancy Mosa, Ph.D.

Academic Editor

PLOS ONE

Journal Requirements:

3. We note that figure 3 in your submission contain map images which may be copyrighted. All PLOS content is published under the Creative Commons Attribution License (CC BY 4.0), which means that the manuscript, images, and Supporting Information files will be freely available online, and any third party is permitted to access, download, copy, distribute, and use these materials in any way, even commercially, with proper attribution. For these reasons, we cannot publish previously copyrighted maps or satellite images created using proprietary data, such as Google software (Google Maps, Street View, and Earth). For more information, see our copyright guidelines: http://journals.plos.org/plosone/s/licenses-and-copyright.

   a. You may seek permission from the original copyright holder of Figure 3 to publish the content specifically under the CC BY 4.0 license.   

Reviewers' comments:

Reviewer's Responses to Questions

**Comments to the Author**

1. Is the manuscript technically sound, and do the data support the conclusions?

Reviewer #1: Yes

Reviewer #2: Yes

Reviewer #3: No

Reviewer #4: Partly

Reviewer #5: Yes

2. Has the statistical analysis been performed appropriately and rigorously? 

Reviewer #1: Yes

Reviewer #2: Yes

Reviewer #3: Yes

Reviewer #4: Yes

Reviewer #5: Yes

3. Have the authors made all data underlying the findings in their manuscript fully available?

Reviewer #1: Yes

Reviewer #2: Yes

Reviewer #3: Yes

Reviewer #4: Yes

Reviewer #5: Yes

4. Is the manuscript presented in an intelligible fashion and written in standard English?

Reviewer #1: Yes

Reviewer #2: Yes

Reviewer #3: Yes

Reviewer #4: Yes

Reviewer #5: Yes

5. Review Comments to the Author

Reviewer #1: The abstract needs to include a few noteworthy findings.

The study is limited to the province of Hunan in China. In what ways, in your opinion, does the geographic focus impact the application of your findings to other regions or countries?

This study utilized traffic accident data collected by the Hunan Provincial Public Security Department in China, covering the 3-year period from 2014 to 2016. In my opinion, the data on this topic is unhelpful because it is outdated. What are the advantages of using it? but Figure 1 shows up to 2021??

The figures have a very low resolution.

Given the age-specific findings, what targeted strategies would you recommend for improving the safety of vulnerable populations like children and the elderly?

How do your findings compare to similar studies conducted in other regions or countries? Are there any significant similarities or differences that you would like to highlight?

Limitations and Future Research. Does this have to be in your article? It's common knowledge that no study can account for every factor.

Reviewer #2: 1. The abstract need more clear about the main finding and conclusions of the research.

2. need to arrange all figures and tables.

3. The study area map needs to add the north arrow and coordination if available. And important to add the legend to define the location of accidents.

4. The regression model developed need validation or verification with field data.

5. What are the main recommendation that the author can advise for alleviate the accidents and reduce risk to pedestrian through utilizing transportation system in general.

6. All figure are not clear

7. All tables are not clear

Reviewer #3: After extensive review of the research submitted for the purpose of evaluation, it was concluded that the research is very good in terms of choosing the topic and the mechanism of presentation and submission, and it can be accepted without fundamental modifications. The evaluator received comments and a simple inquiry, as shown.

1. Since the different safety effects of the various user groups were found, has the effect of the nature of pedestrians been studied in terms of the nature of physical activity and structure of the city referred to, and the extent to which this affects the analysis. The study city is considered one of the active locations in terms of business and the nature of its population.

2. Possibility of adding more keywords.

3. The installed figures are not clear. Please install the figures in higher resolution as in figure 1.

Reviewer #4: This paper studied the influence of road environmental factors on traffic accidents, which can provide reference to the pedestrian safety in the transportation system design. However, some problems need to be addressed. My comments are provided below.

1. The author selected traffic accident data collected by the Hunan Provincial Public Security Department in China from 2014 to 2016. The data seems out of date, and the author needs to add data from recent years (2017-2024) to derive more reliable results.

2. Did the author only consider collisions? What about other accidents, such as fire, explosion, scraping and rolling etc.?

3. The resolution of the figures is too low, and it is difficult for me to understand them. The author is encouraged to provide vectograph.

4. The language of this manuscript needs to be polished by a native speaker.

5. The application value of the article should be more prominent in the Abstract part.

Reviewer #5: Recommendation: major revisions

Comments:

This paper mainly discusses the impact of road environment factors on traffic accidents involving vulnerable road users (children and the elderly). Through the analysis of traffic accident data from Hunan Province, the study uses a negative binomial regression model and a zero-truncated negative binomial model (ZTNB) to assess the effects of factors such as road functional classification, intersection design, traffic control, and isolation facilities on the frequency of accidents. However, there are still some issues:

1.In fact, there are many derived models of negative binomial models, but there are few methods and models in the introduction, especially for the application and advantages of ZTNB model, the description is insufficient.

2.This research is based on traffic accident data from Hunan Province from 2014 to 2016, which covers a limited time span. The authors are advised to discuss whether this time period is generally representative, and to conduct a certain restrictive discussion on the generalizability of the research results.

3.It is suggested that the authors provide a more detailed explanation of the selection of road environment variables, particularly explaining why certain variables were included in the analysis while other potentially important factors (e.g., weather conditions, time factors) were not considered, to ensure the comprehensiveness of the research results.

4.It is recommended to further strengthen the policy recommendations in the conclusion section, clearly specifying which road environment improvement measures are most likely to effectively reduce traffic accidents, and providing specific implementation paths. This will offer more actionable advice for policymakers and urban planners.

5.In the fourth paragraph of Section 5.3.1, there are repeated statements regarding the quantity of each sample.

6. PLOS authors have the option to publish the peer review history of their article (what does this mean?). If published, this will include your full peer review and any attached files.

Reviewer #1: No

Reviewer #2: **Yes: **Dr. Zainab Ahmed Alkaissi

Reviewer #3: **Yes: **Asst.Prof. Dr. Rana Amir Yousif

Reviewer #4: No

Reviewer #5: No

---

## [Author Response · Author response to Decision Letter 0]

15 Nov 2024

Thank you for your efficient work in processing our manuscript (PONE-D-24-31325). Those comments are very beneficial to improve our paper, and they also can be seen as the important guidance to our research. We have studied comments carefully and have made targeted revisions. The changes are marked red in the revised manuscript. Point by point responses to the reviewers’ comments are listed below.

In accordance with the laws and regulations stipulated by the pertinent authorities in China, the confidentiality and privacy of traffic accident data must be upheld and cannot be divulged.

---

## [Decision Letter · Decision Letter 1]

11 Dec 2024

PONE-D-24-31325R1Influence of Road Environmental Factors on Traffic Accidents Involving Vulnerable Road Users through Negative Binomial ModelsPLOS ONE

Dear Dr. Tian,

Thank you for submitting your manuscript to PLOS ONE. After careful consideration, we feel that it has merit but does not fully meet PLOS ONE’s publication criteria as it currently stands. Therefore, we invite you to submit a revised version of the manuscript that addresses the points raised during the review process.

We look forward to receiving your revised manuscript.

Kind regards,

Nik Hisamuddin Nik Ab. Rahman

Academic Editor

PLOS ONE

Additional Editor Comments :

Major revision

Reviewers' comments:

Reviewer's Responses to Questions

**Comments to the Author**

1. If the authors have adequately addressed your comments raised in a previous round of review and you feel that this manuscript is now acceptable for publication, you may indicate that here to bypass the “Comments to the Author” section, enter your conflict of interest statement in the “Confidential to Editor” section, and submit your "Accept" recommendation.

Reviewer #2: All comments have been addressed

Reviewer #3: All comments have been addressed

Reviewer #4: (No Response)

Reviewer #5: All comments have been addressed

2. Is the manuscript technically sound, and do the data support the conclusions?

Reviewer #2: Yes

Reviewer #3: Yes

Reviewer #4: (No Response)

Reviewer #5: Yes

3. Has the statistical analysis been performed appropriately and rigorously? 

Reviewer #2: Yes

Reviewer #3: Yes

Reviewer #4: (No Response)

Reviewer #5: Yes

4. Have the authors made all data underlying the findings in their manuscript fully available?

Reviewer #2: Yes

Reviewer #3: Yes

Reviewer #4: (No Response)

Reviewer #5: Yes

5. Is the manuscript presented in an intelligible fashion and written in standard English?

Reviewer #2: Yes

Reviewer #3: (No Response)

Reviewer #4: (No Response)

Reviewer #5: Yes

6. Review Comments to the Author

Reviewer #2: Need to add only limitation for the work more clearly, And all the required notes has been done

Also, And All the questions has been clarified

Reviewer #3: After extensive review of the research with the symbol ModelsPONE-D-24-31325R1and tagged Influence of Road Environmental Factors on Traffic Accidents Involving Vulnerable Road Users through Negative Binomial, the following was reached:

The reviewer recommends accepting the aforementioned research after observing the following:

1. The researcher made all the modifications required by him and determined by the four evaluators.

2. The researcher answered most of the research inquiries directed by reviewers.

3. Although the study period was proposed, which is the years 2014 to 2016, and although the years are not specific, the analysis was done in good ways and the study can be used for good future research.

With respect.

Reviewer #4: (No Response)

Reviewer #5: The reviewer recommends acceptance for publication.

The author has responded to the reviewers' comments and made corresponding revisions according to their suggestions. There are no further issues.

7. PLOS authors have the option to publish the peer review history of their article (what does this mean?). If published, this will include your full peer review and any attached files.

Reviewer #2: No

Reviewer #3: **Yes: **Asst.Prof.Dr.Rana Amir Yousif

Reviewer #4: No

Reviewer #5: No

---

## [Author Response · Author response to Decision Letter 1]

16 Dec 2024

If there are still parts that need to be improved please contact me and I will actively work on them.

---

## [Editor Report · Decision Letter 2]

2 Jan 2025

Influence of Road Environmental Factors on Traffic Accidents Involving Vulnerable Road Users through Negative Binomial Models

PONE-D-24-31325R2

Dear Dr. Tian,

We’re pleased to inform you that your manuscript has been judged scientifically suitable for publication and will be formally accepted for publication once it meets all outstanding technical requirements.

Kind regards,

Nik Hisamuddin Nik Ab. Rahman

Academic Editor

PLOS ONE

Additional Editor Comments (optional):

Reviewers' comments:

Accepted as all the corrections are carried out

---

## [Editor Report · Acceptance letter]

14 Jan 2025

PONE-D-24-31325R2 

PLOS ONE

Dear Dr. Tian, 

I'm pleased to inform you that your manuscript has been deemed suitable for publication in PLOS ONE. Congratulations! Your manuscript is now being handed over to our production team.

Kind regards, 

on behalf of

Professor Dr Nik Hisamuddin Nik Ab. Rahman 

Academic Editor

PLOS ONE